# Perspectives on Novel Technologies of Processing and Monitoring the Safety and Quality of Prepared Food Products

**DOI:** 10.3390/foods12163052

**Published:** 2023-08-15

**Authors:** Jinjin Huang, Min Zhang, Zhongxiang Fang

**Affiliations:** 1State Key Laboratory of Food Science and Resources, Jiangnan University, Wuxi 214122, China; leisurehuang1108@163.com; 2Jiangsu Province International Joint Laboratory on Fresh Food Smart Processing and Quality Monitoring, Jiangnan University, Wuxi 214122, China; 3China General Chamber of Commerce Key Laboratory on Fresh Food Processing & Preservation, Jiangnan University, Wuxi 214122, China; 4School of Agriculture and Food, The University of Melbourne, Parkville, VIC 3010, Australia; zhongxiang.fang@unimelb.edu.au

**Keywords:** prepared fried rice, food processing, quality monitoring, food safety

## Abstract

With the changes of lifestyles and rapid growth of prepared food industry, prepared fried rice that meets the consumption patterns of contemporary young people has become popular in China. Although prepared fried rice is convenient and nutritious, it has the following concerns in the supply chain: (1) susceptible to contamination by microorganisms; (2) rich in starch and prone to stall; and (3) vegetables in the ingredients have the issues of water loss and discoloration, and meat substances are vulnerable to oxidation and deterioration. As different ingredients are used in prepared fried rice, their food processing and quality monitoring techniques are also different. This paper reviews the key factors that cause changes in the quality of prepared fried rice, and the advantages and limitations of technologies in the processing and monitoring processes. The processing technologies for prepared fried rice include irradiation, high-voltage electric field, microwave, radio frequency, and ohmic heating, while the quality monitoring technologies include Raman spectral imaging, near-infrared spectral imaging, and low-field nuclear magnetic resonance technology. These technologies will serve as the foundation for enhancing the quality and safety of prepared fried rice and are essential to the further development of prepared fried rice in the emerging market.

## 1. Introduction

In recent years, especially after the COVID-19 pandemic, people’s lifestyles have been changed tremendously, including the food consumption pattern. With the popularity of simple household appliances such as refrigerators and microwave ovens, the markets for prepared foods have grown significantly in North America, Europe, and China. In the convenience food sector alone, revenues reached $562.2 billion in 2022, where China contributed a significant proportion ($126.5 billion), and the convenience food industry is predicted to expand at a rate of 6.21% annually [1]. Prepared fried rice is mainly made from rice, which is stir-fried in cooking oil, seasoned with salt and other ingredients (with pre-processed dishes or ingredients such as eggs, seafood, green beans, carrots, sausages, etc.), and then sterilized and packaged for distribution. Consumers only need a simply reheating process before eating. With a more balanced nutritional profile, prepared fried rice caters to young customers who are busy and have no time to cook, and fits a more efficient and healthier lifestyle. The emergence of smart kitchens and future food manufacturing models have produced a new pattern of Chinese cuisine. In 2016, the State Council of China published the opinions to broaden the supply channel of staple food and create a high standardized and high-quality staple food brand [2]. It promoted industrial production of traditional Chinese staple foods and dishes. As a result, many representatively prepared fried rice brands have emerged in China markets (Table 1). Moreover, the cold chain logistics market of China has maintained an annual growth rate of more than 11% from 2016. The improvement of cold-chain logistics level and the development of food process technologies (including quick-freezing and preservation technologies) provided technical supports for the long-distance supply of prepared fried rice [3]. Due to different food ingredients in making prepared fried rice, appropriate processing and preservation and quality monitoring techniques should be applied to ensure food safety and quality [4]. Most companies currently choose −18 °C for low-temperature preservation during transportation. Convenience and health, low carbon emissions are the most critical development direction of prepared foods worldwide. Some companies are already experimenting with 4 °C refrigerated preservation with a view to reducing cold-chain transportation costs.

Prepared fried rice is a type of emerging food with Chinese characteristics (taste, flavor etc.), and there are few reported pieces of research on it. Similar to Chinese fried rice, Thailand has rice cracker snacks that require deep-frying [5], and Korea has the traditional oil-puffed snack Yukwa [6,7]. Although there are representative brands of fried rice in Japan, there is no classification of the types of fried rice. Most European and American consumers are used to eating risotto. Research has mainly focused on the pathogenicity [8] and growth kinetics [9,10] of harmful microorganisms in fried rice, with little discussion on the potential applications of novel physical processing technologies and nondestructive testing techniques on prepared fried rice. The processing of prepared fried rice is divided into the treatment of raw materials before cooking; the way of flipping during cooking; and the sterilization, packaging, retarding aging, and reheating after cooking. Uneven reheating can affect the flavor and texture of prepared fried rice. Therefore, the post-cooking processing technology is critical. Traditional heat treatment sterilization techniques (pasteurization: temperature below 100 °C; high temperature sterilization: temperature higher than 100 °C; ultra-high temperature instant sterilization: temperature at 135~150 °C) could destroy the flavor substances and color of the prepared fried rice. Physical field technology and anti-bacterial coating technology consume less energy and can retain the inherent nutrients and freshness of raw materials, thus minimizing the deterioration of quality caused by conventional sterilization processes. Moreover, physical field technologies have wide applications in enzymatic reaction [11], fermentation [12], and retarding aging [13]. Their advantages of no additives and continuous inhibition of harmful microbial growth and reproduction are more readily accepted by consumers. Therefore, this paper focuses on the potential sterilization technologies (irradiation, high-voltage electric field, microwave, radio frequency, ohmic heating), and quality monitoring (Raman spectral imaging, near infrared spectral imaging, and low-field nuclear magnetic resonance) technologies for prepared fried rice (Figure 1). These technologies are crucial to ensure the quality and safety of prepared fried rice and will provide guidance for further research and development for this group of emerging food products.

## 2. Potential Processing Technologies for Prepared Fried Rice

### 2.1. Sterilization Technologies

Unfavorable conditions during prepared fried-rice processing or storage can lead to the growth of pathogenic bacteria and hazardous substances produced by microbial metabolism [9]. As a raw material, rice can become contaminated during cultivation and storage by dust, water, plants, insects, soil, fertilizers, and animal dung, carrying harmful substances such as aflatoxin and *Bacillus cereus* [14]. There are potential biohazards associated with the use of contaminated rice for fried rice preparation. Moreover, the processing methods, factory sanitary conditions, packaging methods, and storage environment also affect the quality and safety of prepared fried rice [9]. *Vibrio parahaemolyticus* had a high growth rate in egg fried rice at room temperature due to inadequate sterilization and cold chain systems [15]. In prepared fried rice, microbial contamination mainly appeared on the surface of fried rice. Heat-sensitive foods such as egg whites or prepared fried rice were prone to contain *Bacillus cereus*, which is highly resistant to high temperatures and acidity during cooking and may cause acute liver failure and encephalopathy if ingested [16]. Typically, rapid cooling is applied after the fried rice is cooked, but it is also the most vulnerable period to microbial contamination. The correct refrigeration temperature was used during storage to control the harmful microorganisms, but those with spores remained alive [17]. *Bacillus cereus* may grow exponentially in fried rice when cooked in ham fried rice, scrambled egg fried rice, and pea fried rice, which were slowly cooled and stored at 15 °C [18]. Diarrhea and vomiting in accidental eaters of fried rice in a Korean restaurant were reported, due to the contamination of *Bacillus cereus* with a total colony count of 1.48–3.47 log CFU/g, presumably due to spoilage from improper storage [19]. In addition, eggs and eggshells were susceptible to *Salmonella* contamination, and *Salmonella* serovar can cause severe gastrointestinal disease [20]. Some researchers have studied the growth and hemolysin production mechanism of two strains of *Vibrio alginolyticus* and reported their toxicity in fried rice to raise public awareness of the safety of fried rice [21]. The examples shown above demonstrated a high possibility of microbiological contamination in prepared fried rice. Prepared fried rice should be primarily inhibited from harmful microorganisms such as *Bacillus cereus*, *Vibrio parahaemolyticus*, and aflatoxins produced by *Aspergillus flavus* and *Aspergillus parasiticus* during processing. Ensuring fresh raw materials and good sanitary environment during processing are critical to the safety and quality of prepared fried rice. The following are some important methods that could be applied to further improve the product safety and quality.

#### 2.1.1. Irradiation

Irradiation is a non-thermal method that is frequently used to preserve food, where food is subjected to a specific level of ionizing radiation generated by mechanical or natural sources; and the DNA of microorganisms is destroyed, cell division is prevented, and the cells stop activities, thus providing a sterilizing effect. Generally, electron beam, X-ray, and γ-ray are used for food irradiation. Mechanical sources produce X-rays and electron beam, which can be turned on and off. γ-rays are generated by radionuclides ^60^Co and ^137^Cs. They have different effects on microorganisms, with penetration depths of about 8 cm, 20 cm, and 40 cm, respectively [1]. However, low consumer acceptance to irradiated food is an issue for the rapid application of this technology, although the Joint Expert Committee on Food Irradiation of the World Health Organization has demonstrated the safety of food irradiation at doses below 10 kGy and that these sources are not radioactive to cause safety hazards [22]. Moreover, appropriate irradiation doses are beneficial to preserve the color and nutrition of the food. In an investigation into the shelf-life of ready-to-eat fried diced chicken with chili, researchers found that 10 kGy γ-rays effectively killed pathogenic bacteria, such as *Salmonella*, *Escherichia coli*, *Bacillus* spp., and *Staphylococci*, with no loss of sensory quality and protein content [23]. Prepared fried rice usually contains rice, meat, and green vegetables, such as green beans and cabbage. It is crucial to take into account how the sterilization methods will affect the protein, fat, and starch in the fried rice and reduce its impact on the flavor and quality of the rice and vegetables during the sterilization process. Researchers found that irradiating brown rice with an intensity of 1 kGy γ-rays dramatically reduced its microbial population (molds and yeasts) after two-month storage without affecting the overall sensory tolerability. The irradiated samples became softer and the members of the sensory evaluation panel considered that the cooked brown rice was of better quality under 1 kGy irradiation [24]. Furthermore, a dose of 1 kGy γ-rays was discovered to be adequate to significantly lower the level of *Escherichia coli* (ATCC# 25922) in all mixed vegetables and increase the shelf life by up to 4 days [24].

Although irradiation benefits from a quick processing period and environmental friendliness, it also has limitations. For electron beam, the energy and dose limit are 10 MeV and 10 kGy. The maximum energy and dosage for X-rays are 5 MeV (US 7.5 MeV) and 10 kGy. The maximum energy and dosage for γ-rays is 5 MeV and 10 kGy [1]. High doses (4.5 kGy) of electron beam irradiation intensity can oxidize lipids and vitamins in foods [25]. Moreover, it has been shown that irradiation levels of up to 44 kGy are necessary to inactivate microbial spores, but the current commonly used 10 kGy is far below this level. Therefore, it is necessary to combine irradiation with other techniques to achieve effective bactericidal effects [26]. For example, combining different doses of γ-irradiation with 4 °C cold storage significantly reduced the number of toxic *Bacillus cereus* [27]. γ-irradiation combined with active coating at 0.4 kGy had a synergistic inhibition effect on *Escherichia coli* and *Aspergillus niger* [28]. Therefore, combining radiation with other techniques could be an effective method in the preservation of prepared fried rice.

#### 2.1.2. High-Voltage Electric Field

High-voltage electric field sterilization technology is an effective, low energy-consuming, and non-polluting method that can preserve food quality. The high-voltage electric field is divided into high-voltage pulse electric field and high-voltage electrostatic field. High-voltage pulsed electric fields apply a strong electric field to food within a relatively short time. Compared to monopolar pulses, bipolar pulses have a greater impact on the porosity of cell membranes [29], and increasing the number of pulses and pulse frequency can also enhance the sterilization effect [30]. High-voltage pulsed electric field sterilization can be carried out at room temperature. It causes damage to the bacterium’s enzymatic activity, DNA, and protein structure, impairing its typical physiological function and resulting in cellular breakage or death [31]. Prepared fried rice with ingredients including rice, green beans, and carrots may have been infected with *Staphylococcus aureus* and *Bacillus licheniformis* on the farm or during production. High-voltage pulsed electric fields have significant lethal effects on common foodborne pathogens such as *Staphylococcus aureus*, *Salmonella typhimurium*, *Escherichia coli*, and *Listeria monocytogenes* [32]. Fernandez-Diaz et al. (2000) concluded that pasteurization at 55.6 °C for 6.2 min or 56.7 °C for 3.5 min may not inactivate *Salmonella* spp. and *Listeria monocytogenes* [33]. While applying 26 kV/cm and 37 °C with 100 exponential decay pulses of 4 μs to continuously processed the liquid eggs, the proteins do not coagulate and the flavor substances of the eggs are retained; however, the harmful microorganisms decrease with the number and width of pulses increasing. They can be combined with other techniques to obtain better bactericidal effects. When high-voltage pulse electric field technology and cryoconcentration are used together, sublethal microbes can be suppressed without causing heat damage to the food [34]. Similarly, with the addition of nisin, a synergistic effect occurred and the bactericidal effects of the pulsed electric field was more effective [33].

High-voltage electrostatic field equipment is relatively simpler and lower cost compared with that of a high-voltage pulsed electric field (Figure 2). The electric field strength is also lower. High-voltage electrostatic fields kill mold and bacteria on food surfaces by causing small amounts of charged particles present in the air to constantly collide with neutral molecules or atoms. High-voltage electrostatic fields also cause oxidative stress and DNA sub-damage in bacteria, which have a sterilization effect on both solid and liquid foods, reducing the number of chemical preservatives and preserving the nutrients of the food [35]. Qi et al. (2021) studied the bactericidal effect of high-voltage electrostatic fields on salmon, pork, and sausage, in which the lethality of *Staphylococcus aureus* reached 92.1–99.8% [36]. These ingredients are often used in different types of prepared fried rice. Hsieh et al. (2011) measured the sensory quality index of tilapia at an electric field intensity of 1 kV/cm, and the treated group was fresher compared with the untreated ones, where the electric field intensity inhibited the protein denaturation of fish and significantly reduced the number of microorganisms [37]. Ko et al. (2016) obtained the same conclusions. After successive treatments with high voltage electrostatic fields at 3, 6, and 9 kV/cm for 8 days, it maintained the protein content, solubility, and Ca_2_-ATPase activity [38]. *Acinetobacter johnsonii*, which is commonly existed in the water and soil around fish culture, is also present in fresh or spoiled processed foods and possibly in prepared fried rice with complex ingredients. After 15 min of treatment with a 30 kV high-voltage electrostatic field, the number of *Acinetobacter johnsonii* decreased, intracellular nucleic acids and proteins leaked, conductivity and reactive oxygen species (ROS) content increased 16.88-fold [39]. The sterilizing effect can be improved by combining high-pressure electrostatic field technology with other technologies. For example, the combination of high-pressure electrostatic field and controlled freezing point technique (temperature range from 0 °C to the freezing temperature of the organism) can successfully suppress microbial development and reproduction and postpone the loss of food quality [38].

The prepared fried rice needs to be reheated before consumption, i.e., the second heating of the cooked food. Although reheating prepared fried rice is a common consumer behavior, many restaurants are now introducing prepared fried rice to improve the efficiency of meal delivery, and convenience stores are selling prepared fried rice and providing reheating services. Therefore, new processing technology of reheating should be taken into consideration for prepared fried rice. Reheating can cause changes in sensory quality such as moisture, texture, color, starch pasting, digestibility, and other physical and chemical properties of food. The current research on reheating technologies focuses on the overcooked taste and the food quality after reheating. For prepared fried rice, moving closer to the quality of restaurant fried rice after reheating has attracted more interest. Traditional reheating, such as steaming, can maintain certain qualities of the food, but has the disadvantage of taking a long time and adding moisture to the food. Recently, microwave reheating and radio frequency reheating have become popular in reheating processing.

#### 2.1.3. Microwave

Microwave heating has both a sterilization and reheating effect on the food. Microwave-assisted pasteurization is the process of pasteurization by using microwaves to act directly on the material, using the movement of the food’s own polar molecules and conductive ions, friction, to generate heat. Compared to traditional heating methods, microwave heating is three to five times faster [40]. In addition, it reduces the loss of nutrients, flavor substances, and texture deterioration during food heating [41]; reduces thermal degradation of heat-sensitive ingredients; improves food quality; and extends shelf life [42]. For instance, *Bacillus cereus* was inoculated in fried rice and then heated in a frying pan at high temperature (internal temperature of fried rice: 103.8–121.4 °C), medium temperature (internal temperature of fried rice: 69.2°C), or microwave (internal temperature of fried rice: 86.3–90.6 °C) for 3 min, respectively; and the results showed that heating in microwave oven was the most effective to control the *Bacillus cereus* cells and spores. With microwave heating, the fried rice can be kept safe at 25 °C for 6 h, 35 °C for 3 h, and 45 °C for 2 h [43]. After microwave-assisted pasteurization, Montero et al. (2020) examined the physical, chemical, sensory, and microbiological properties of fried egg rice. The treated product had a shelf life of up to 6 weeks at 7 °C storage, where no sensory related microbial-induced spoilage was detected; and the eggs had an acceptable firm texture, showing the great potential of microwave-assisted pasteurization for this group of food [44]. It is noteworthy that 10% of American consumers store their food at a temperature of 7 °C most of the time [42], indicating the practical application of this technology.

Microwave not only has a good sterilization effect, but also a reheating effect. When microwave applies, the whole food will be heated up instantly and homogeneously. Researchers have investigated the quality loss and flavor characteristics of food products during microwave reheating. It was reported that meat products have little difference in cooking loss during microwave reheating compared to other reheating methods such as steaming, because of microwave’s high heating efficiency [45]. Microwave can also well-retain active ingredients such as ascorbic acid in vegetables; and the loss of total phenols, total anthocyanins, and chlorogenic acid is much lower than that of stir-frying and deep-frying [46,47]. However, it is important to note that hot and cold spots caused by uneven microwave heating and moisture loss can lead to bland sensory quality of foods after microwave reheating. Geedipalli, Rakesh, and Datta (2007) placed 3.6 × 4.7 × 2.1 cm potato cubes in the center of a turntable inside a 2459 Hz microwave oven and heated for 35 s by completing a rotational cycle every 10 s [48]. Figure 3 compared the effect of the same oven with and without the turntable on the uniformity of potato temperature distribution. The static-heated potatoes had concentrated high-temperature areas, while the potatoes heated by the combined microwave rotary heating had more uniform temperature contours on their surfaces and were heated more efficiently with low-quality loss. Moreover, the electric field applied to the microwave cavity and temperature distribution can be influenced by the food dielectric characteristics and microwave frequency [49,50,51,52]. Therefore, choosing the appropriate microwave time and power is important to achieve the best results for reheating prepared foods. Researchers compared the sensory score and temperature distribution of lasagna with different combinations of frying time (5.5, 7, and 8.5 min) and microwave reheating power (low, medium low, medium, medium high, and high power), and determined that a frying time of 7.0 min and microwave power of medium-low power (4.20 W/g) and 80 s reheating resulted in the best quality. Wang et al. (2018) studied the uneven temperature distribution of convenience rice during microwave reheating and proposed an intermittent reheating power and packaging container design method to improve the microwave heating uniformity [53]. In addition, combining microwave and infrared technology was considered more effective than the single microwave heating [54]. This is because polar molecules and conductive ions inside the dish produced heat as a result of microwave; and infrared tubes radiate heated the food from the external environment to reduce the temperature difference between the interior and exterior, therefore inhibiting the migration of polar molecules such as moisture and enhancing temperature uniformity [55]. Lasagna, a pre-made pasta, is similar to prepared fried rice. It is made in a similar process and the ingredients are rich in starch, with a supplementary of oil, salt, and others; in addition, both need to be kept refrigerated [56]. The way lasagna is reheated can also be considered as a potential prepared fried rice reheating method. Comparing traditional reheating methods (steam, baking) and new reheating methods (air fryer, microwave, and combined infrared microwave), lasagna reheated by infrared combined with microwave had better temperature uniformity than microwave alone and increased internal moisture to 33.85%. This method also inhibited the outward migration of moisture, reduced the hardness, and improved the sensory property [57].

#### 2.1.4. Radio Frequency

Radio frequency (RF) is a high-frequency alternating current electromagnetic wave with frequencies between 300 kHz~300 MHz; and only 13.56 MHz, 27.12 MHz, and 40.68 MHz are used in industry [4]. RF treatment not only has a sterilizing effect on the individual components of prepared fried rice, but also has the effect of reheating and regulating the structure, digestibility, and physicochemical properties of rice starch [58].

In terms of the sterilization effect, RF penetrates the material interior and causes the rhythmic migration of charged ions, converting electrical energy into heat, and rapidly increasing the bulk temperature of the prepared fried rice. With a deep penetration rate and high energy efficiency, it overcomes the defects of long traditional heating time and excessive heating at the edges of packaged food [59]. The RF sterilization process is milder, but the effect is similar with the conventional retorting method; in addition, its effect on the quality deterioration of the product, such as nutrient content, texture, taste, and odor, is significantly reduced. RF has a high potential to be applied for prepared fried rice processing. It was observed that freshly cooked eggs were better preserved with a significantly longer shelf life after RF sterilization, so the U.S. military used RF technology to improve the shelf life of eggs to meet soldiers’ needs [60]. After the sterilization of *Nostoc sphaeroides* using 6 kW, 27 MHz RF for 20 min, microbial counts were significantly reduced and the effect on its color and flavor was significantly lower than that of conventional autoclaving [61]. In addition, researchers have conducted extensive research on the sterilization effect of RF in combination with other techniques on food products. It was found that the effects of ZnO nanoparticles combined with RF heating on the sterilization of carrots and product quality characteristics was superior to either ZnO nanoparticles or RF heating treatment alone and increased the shelf life of prepared carrots up to 60 days [62].

For the research on the reheating effect of RF, most previous experiments only sterilized or reheated one ingredient; and it is difficult to evaluate the effect on prepared fried rice, as it has multiple ingredients. There are few studies using RF to examine the reheating behavior of non-homogeneous food materials. Lan et al. (2020) performed RF reheating of pizza, which is also a heterogeneous food. With selected RF parameters of 6 kW, 27.12 MHz, a maximum reheating temperature of 85 °C, and an electrode gap of 110 mm, a good consistency of pizza bases, cheese, salami, and onions temperatures after 7.33 min of reheating was obtained. The results suggested that heating non-homogeneous foods with similar loss factors and dielectric constants may require higher RF heating rates and that the uniformity of RF heating can be achieved by the moving or rotating of the raw materials [63]. Wang et al. (2012) also reheated meat lasagna noodles, a kind of highly heterogeneous food, with RF. When the beef, mozzarella cheese, noodles, and sauce were properly distributed, they showed little difference in temperature and maintained good product quality after the reheating process [64].

#### 2.1.5. Ohmic Heating

Ohmic heating is also known as energized heating, where the food is acted as a resistance unit and electrical energy is applied through two electrodes located on both sides of the heating chamber in direct contact with the food material (Figure 4). When an electric current is passed through a conductive material, the internal temperature of the material increases due to the conversion of electrical energy into thermal energy by the Joule effect. Although convective heat-transfer modes are still the most popular heating technologies in the food industry, ohmic heating with its time and energy intensive advantages could serve as a potential alternative to traditional methods in prepared fried rice [65]. Ohmic heating is used in food industries for sterilization, thawing, extraction, and others [66], which has benefits of a high rate of electric heat conversion, simple and easy-to-control equipment, and no scaling, etc. [67]. It can save heating time by 48% and is more energy efficient than steam heating. Additionally, unlike microwave heating, there is no need for radiation isolation material during processing to protect the operator [68]. The effectiveness of ohmic heating is directly impacted by the electrical conductivity of the food substance. Electric field strength is proportional to the square of electrical conductivity; however, as particle size rises, electrical conductivity decreases [69]. As shown by Benabderrahmane and Pain (2000) and Icier and Ilicali (2005), conductivity decreases with increasing stacking density [70,71]. After salting, the electrical conductivity of the food material increases [72]. The conductivity is also influenced by particle position and orientation, with the solid phase heating up more quickly than the liquid phase under series settings and the opposite being seen under parallel conditions [69]. The material state of the food, the contact area between the electrodes and the food material, and the distance between these electrodes also affect the heating rate.

Ohmic heating has a non-thermal effect (50–60 Hz) due to the presence of an electric field. The cell membrane has a specific dielectric strength and the presence of an electric field exceeding the electric field strength leads to changes in the cellular tissue structure. The cell wall accumulates charge and forms pores, cell membrane permeability is enhanced, and intracellular components diffuse, thus inactivating microorganisms [73]. Ohmic heating has a bactericidal effect on *Bacillus licheniformis*, *Escherichia coli*, *Bacillus subtilis*, and *Streptococcus thermophilus* that may be present during the processing of prepared fried rice [74]. It can even be used to inactivate heat-resistant bacterial spores, such as the most heat-resistant *Bacillus thermophilus* spores [75]. However, Somavat et al. (2012) argued that additional non-thermal effects can only be observed under specific treatment conditions or in specific food samples [75]. If a stronger non-thermal effect in prepared fried rice is required, its electrical conductivity should be increased, for example, by changing the stacking density of the rice grains. The thermal effects frequently conceal the non-thermal effects of ohmic heating that are used to inactivate foodborne bacteria. However, when other methods are utilized in addition to ohmic heating, the non-thermal effect is more pronounced and the pathogen inactivation rate rises dramatically. Combined treatments of non-homogenized salsa (tomatoes, onions, small peppers, salt, black pepper powder, lemon juice) with ohmic heating and parsley phenol had a synergistic bactericidal effect on foodborne pathogens such as *Escherichia coli O157:H7*, *Salmonella Typhimurium*, and *Listeria monocytogenes*. Compared to single ohmic heat sterilization, the combined treatment is less destructive to the composition of the food and does not affect the color and quality of the salsa [76]. Similarly, the combination of ohmic heating with citral or muscimol showed a synergistic bactericidal effect against the above-mentioned food-borne microorganisms [77]. Because foodborne pathogens can survive for long periods in cold spots, De Alwis and Fryer (1990) applied a mathematical modeling approach to simulate the ohmic heating process to improve food safety [78]. Choi et al. (2020) also applied mathematical modeling methods to accurately simulate the temperature distribution and inactivation of *Escherichia coli O157:H7*, resulting in a 5-log reduction [79].

Normally, thermal processing affects the texture of food, causing water loss and hardening. Ohmic heating reduces the damage to food quality. Compared to traditional steaming methods, ohmic heating consumes less energy and accelerated softening of white rice, brown rice, and germinated brown rice. It also offers greater advantages by reducing the loss of hardness by 50% after the pretreatment of foods containing starch [66]. These characteristics of ohmic heating are extremely important in the reheating of prepared fried rice. Combining ohmic heating with microwave and Near infrared radiation can improve the uniformity of heating [80]. The combination of cold plasma, RF and ohmic heating significantly reduces the processing time and the target temperature for ohmic heating. By using combined heating, food quality degradation can also be minimized and food safety improved. The elderly have special needs for food nutrient content, and processing techniques should maximize the retention of the nutrients and flavor. Joe et al. (2021) developed a combined ohmic and vacuum heating system to process elderly foods. The effects of vacuum pressure intensity and action time on the physical and electrical properties of foods with solid–liquid mixtures were discussed. It was found that the combined technique shortened the heating time of foods and solved the problem of uneven temperature distribution [81]. This provides an idea for the processing of prepared fried rice. Prepared fried rice is a kind of heterogeneous food, in which the fat wraps around the rice grains and auxiliary ingredients to slow down the aging of starch and reduce the viscosity change of fried rice.

At present, ohmic heating technology is still in its infancy and has some challenges. If electrode materials are not selected properly, ohmic heating may suffer from the electrode corrosion of toxic chemicals and overheating of the foods. Ohmic heating requires corrosion-resistant, non-polluting electrodes, and controlled heating speed to avoid local overheating [82]. In addition, for non-homogeneous foods such as prepared fried rice, the particle density affects the heating effect. The energization should be combined with other techniques to avoid uneven currents due to different resistances of the components inside the prepared fried rice. Since the electrical conductivity of the mixture of rice and water is low, it has been suggested that salt can be added to the rice mixture to increase the electrical conductivity [72]. It has also been suggested that soaking rice before making fried rice can effectively increase the electrical conductivity of the mixture, thus making it possible to steam rice with ohmic heating. It is noted that ohmic heating has been rarely reported in foods containing fat and oil, and further research is needed.

## 3. Potential Quality and Safety Monitoring Technologies for Prepared Fried Rice

Most of the traditional techniques used to analyze harmful chemical components and contaminants in food such as high-performance liquid chromatography (HPLC), gas chromatography (GC), and HPLC or GC combined with mass spectrometry are expensive, time-consuming, and invasive to the samples, requiring specialized technician operating [83]. Non-destructive, portable, and environmentally friendly Raman spectroscopy [84], near-infrared spectroscopic imaging [85], and low-field nuclear magnetic resonance [86] have been developed rapidly in recent years and have the potential for rapid and efficient analysis in the field of food quality monitoring. In particular, the combination of chemometric methods for predicting food quality has received wide attention [87]. However, there are few literature reviews on the quality monitoring techniques for prepared fried rice.

Prepared fried rice is rich in proteins and fats, which are highly susceptible to microbial attack and lead to food spoilage. Microorganisms convert nitrates to nitrites in prepared fried rice with Chinese sausage. Fried rice grains are rich in carbohydrates; and after cooking at high temperatures, free amino acids (asparagine) in the food react with reducing sugars or other carboxyl compounds to produce acrylamide [88]. Both nitrites and acrylamide are harmful to human health [89]. Food additives may also undergo transformation and produce toxins when interacting with other ingredients in the food or during heat treatment and freezing. Prepared fried rice packaging materials in an oil and salt environment may also migrate some plastic components into the food, creating potential food safety problems [90]. Therefore, the use of fast and effective detection techniques can reduce harmful substances in these food products.

### 3.1. Raman Spectroscopy Imaging Technology

Raman spectroscopy is an analytical technique used in the study of molecular structure to determine the vibration and rotation of molecules by examining the scattering spectra of various frequencies of incident light. Raman spectroscopy imaging is a combination of Raman spectroscopy and imaging methods that can simultaneously acquire structural morphological data and distribution images of the prepared fried rice. Different hazardous substances may have different spectral characteristics. After obtaining the Raman spectral data of prepared fried rice, the data can be analyzed and interpreted by using professional software. By comparing the spectral profiles of known hazardous substances, the presence of hazardous substances in fried rice can be determined [82]. Prepared fried rice needs to be stir-fried in oil and may contain about 500 µg kg^−1^ of acrylamide [91]. Insertion of the aggregating agent into the metal nanoparticle aggregates to generate more hotspots, which theoretically produces high SERS surface resonance to further enhance the Raman signal [92]. Cheng et al. (2019), combining QuECHERS extraction and SERS, and Ye et al. (2023), using AgNP substrate-assisted SERS technique with 0.5 M NaCl as the agglomerating agent, both demonstrated that Raman spectroscopy can rapidly detect acrylamide in fried foods; and the results are consistent with LC-MS/MS, which can be used as a technical tool for on-site screening [92,93]. It is worth noting that the SERS technique has limitations. The experimental conditions and control parameters need to be optimized because of various samples with different surface structure and preparation. It is currently only used in typical fried foods such as potato chips, needing to further expand the scope of application. Raman spectroscopy techniques can also provide data on the spatial structure of proteins based on the backbone structure and side-chain microenvironment of the protein polypeptide, as well as reliable identification and imaging of starch and gluten [94]. According to the different characteristic peaks corresponding to different structural information, starch molecular structure, protein and fat composition, and additives content were predicted by comparing the variations in the Raman spectra of the prepared fried rice. This information is necessary to evaluate the prepared fried rice quality and safety as well as to detect microorganisms on food surfaces [95]. This technique does not require the complex pretreatment of the prepared fried rice; it is simple to operate and combines with fingerprinting to simultaneously identify multiple molecules for effectively regulating the safety of fried rice ingredients [96], such as rapid online monitoring or on-site detection of pork ractopamine and clenbuterol hydrochloride residues using SERS with chemometric methods [97], non-destructive detection of fungal spore counts and textural features of corn [98], distinguishing of aflatoxin contaminated corn kernels [99], and classification of the fake and real eggs (Figure 5) [100]. These are all raw materials of prepared fried rice. In addition, Raman spectroscopy was used to detect mycotoxins, metal ions, and drug residues in food products (Table 2) [101]. Although the direct application of Raman spectroscopy detection technology on fried rice is not widely reported, its accuracy, rapidity, and wide detection range have great potential for monitoring the quality of this group of foods.

Several companies and institutions such as Thermo Fisher Scientific and HORIBA Scientific have established Raman spectroscopy databases to compare unknown molecules in food products. Dias, Jussiani, and Appoloni (2019) established a database using Raman spectra of 78 major commercial pesticides. In the analysis of pesticides, they used a portable delta nu Raman spectrometer with a laser wavelength of 785 nm and a resolution of 8 cm^−1^ to rapidly detect the pesticide species [109]. The establishment of the database provides a reference for the detection of foods like prepared fried rice that are susceptible to pesticide or microbial contamination.

### 3.2. Near-Infrared Spectroscopy Imaging Technology

The electromagnetic spectrum in the near-infrared (NIR) range (780–2526 nm) is used for spectral analysis. Most commercially available NIR spectrometers are focused on the short-wavelength region of 750–1100 nm [110], while NIR spectral wavelengths in the range of 1100 nm and 2400 nm are associated with branched starch and can be analyzed for starch aging [111]. NIR causes the fried rice molecules to absorb some of the NIR light and transition, reducing the intensity of the light due to different functional groups containing different chemical bonds (O-H, C-H, N-H, S-H bonds, etc.) and hydrogen-containing groups corresponding to different NIR absorption peaks. It is worth noting that this technique is mainly used for macronutrient analysis, of which the content is generally 0.1% higher than that of the sample weight. For prepared fried rice, NIR spectra are not a simple superposition of the individual spectra of each component [101]. Using chemometrics to extract the correct information from the NIR spectra allows qualitative and quantitative detection of water, proteins, and lipids in the pre-made fried rice because of their different structure and composition [112].

The quality of rice decreases with storage time, but it is difficult to quickly distinguish aged rice from the fresh one. Using the advantages of nondestructive and rapid detection by NIR spectroscopy, Shi et al. (2023) combined partial least-squares-discriminant analysis (PLS-DA), support vector machines (SVM), and classification regression trees (CART) algorithms to distinguish the freshness of rice; and obtained that PLS-DA and SVM have excellent classification ability and sensitivity [113]. This is similar to a design of Lapchareonsuk and Sirisomboon (2015) and Onmankhong and Sirisomboon (2021), who proposed to build mathematical models by NIR spectroscopy for quality control in the rice industry [114,115]. Before shipping, prepared fried rice needs to be frozen to reduce the moisture. To detect changes in the quality of pre-made fried rice after freezing, it is usually necessary to measure ice crystal size, cell microstructure, and nutrient content, which is complicated and time-consuming using conventional methods [116]. Since it is technically non-destructive, inexpensive, and real-time, NIR spectroscopy detection has the potential to be used by intelligent applications to quickly identify freezing parameters in food goods and track the freezing process [117]. It is feasible to combine NIR imaging with machine learning to predict the quality of frozen foods because the physical properties of the material components are strongly correlated with the chemical composition and can be reflected by NIR spectral images. This provides ideas for the intelligent monitoring of quality changes in prepared fried rice and predicting shelf life. Jiang et al. (2023) demonstrated the potential of combining NIR spectroscopy with BP-ANN modeling (Figure 6) for the rapid non-destructive detection of frozen foods by modeling the relationship between NIR spectra and quality indicators of frozen samples after thawing using PCR, PLSR, SVR, and a back-propagation artificial neural network (BP-ANN) [118]. In another study, researchers used NIR spectroscopy to determine the fat, moisture, and protein content of frozen beef online during transportation; and this technique allows better control of the production process, resulting in a uniform and consistent quality [119]. There is potential to use this technology for the real-time monitoring of prepared fried rice during the frozen storage. NIR spectroscopy enables the reliable assessment of protein and fat content, and the detection of pH values of beef [120], TVBN, and pH in pork [121]; and moisture and pH in chicken breast [122]. Peyvasteh et al. (2020) used NIR coupled with PCA to determine the relative absorbance changes of oxidized myosin and myoglobin, fat, water, and collagen in the visible spectrum of pork for the rapid evaluation of pork freshness [123]. These ingredients are common sources of protein in prepared fried rice and directly affect its quality.

Currently, the applications of NIR spectroscopy are mostly limited due to the insensitivity of it to impurities. In addition, in dynamic online detection mode, the stability of the instrument and the repeatability of the data are poor and the measurement accuracy is susceptible to environmental factors such as temperature [124]. Therefore, the ability of the models or software to process NIR data need to be further improved to increase the prediction accuracy.

### 3.3. Low-Field Nuclear Magnetic Resonance Technology

Low-field nuclear magnetic resonance (LF-NMR) technology uses atomic nuclei with fixed magnetic moments that exchange energy with alternating magnetic fields in the presence of static and alternating magnetic fields [125]. It is an emerging technology in the field of food quality testing, which is fast, efficient, and non-destructive. Materials of prepared fried rice have different internal compositions and physical properties. The quality classification can be achieved by transverse relaxation time T_2_ and signal intensity. The peaks of LF-NMR spectra are T_21_, T_22_, and T_23_ from left to right, which represent strongly bound water, weakly bound water, and free water in order; in addition, the changes of the position and area of the characteristic peaks reflect the changes of moisture state of the prepared fried rice after processing [126]. This technique has been used in the determination of water content in fruits and vegetables, oil content in plant seeds [127], and water and oil content in fried starches [128].

The quality of prepared fried rice is affected by the temperature fluctuations during storage. High temperatures can damage the resistant starch as well as increase the water absorption of the rice grains, resulting in increased volume, reduced cohesion and adhesion, and floppier rice grains [129]. Prepared fried rice loses moisture content after freezing, thawing, and reheating. After repeated freezing and thawing, the water-holding capacity of food components in the fried rice were affected [130], which can be monitored by the LF-NMR technique. For example, Li, et al. (2012) obtained LF-NMR parameters under different conditions; and determined the color, shear, water retention, pH, and related steaming indicators, concluding that LF-NMR and color measurements are good methods to distinguish the water-holding capacity [131](Li et al., 2012). Ali et al. (2015) reported the analysis of the water state using LF-NMR and showed that after repeated freeze–thaw cycles, lipid and protein oxidation increased with the increasing number of cycles and affected the water-holding capacity [130]. Zhang, Zhang, and Mujumdar (2021) concluded that ice crystals formed after freezing cause damage to the structure of muscle cells, leading to the migration of water molecules. Water loss in chicken breast, beef and seafood, lipid types on fermented sausage and myofibrillar protein gels can be sensitively detected by LF-NMR technique [112]. Moreover, the integrity of proteins is revealed by the decrease in the lateral relaxation time of their protons due to protein aggregation, which further leads to the decrease in the lateral relaxation time of water protons. In addition to monitoring the state and distribution of water, LF-NMR is also effective in the study of detecting the quality of vegetables after water loss. Sun, Zhang, and Yang (2019) studied the trends of NMR signals (A_21_, A_22_, A_23_, and total) of vegetables at different drying powers (200, 300, and 400 W) to obtain the characteristic variables, and used the BP-ANN model for the nondestructive testing of the nutritional quality [132]. Similarly, Zhao et al. (2022) established a method for the non-destructive detection of free-fatty-acid content in frying oil samples based on LF-NMR and BP-ANN [133]. The water and oil contents of prepared fried rice samples have now been examined in the laboratory by LF-NMR. The signals of oil and water were clearly differentiated. The LF-NMR approach is more precise and quicker than the Soxhlet method for the identification of oil-rich fried rice [128].

Interestingly, LF-NMR also played a role in the identification of adulteration in vegetable oils. LF-NMR detection revealed that the imaging difference between vegetable oils and those with additions was the distinction in the transverse relaxation distribution (T_2_ distribution) of the third peak (A) and the shift of the peak T_2_ value, which is likely to be the characteristic peak of polymorphs produced during the frying process of the oils. The A peak area grew linearly as a percentage of the total peak area as the addition of other materials increased. Based on the change in peak area, a linear equation can be established to discriminate adulterations [134]. In another investigation, LF-NMR detected contamination with various hydrocolloids in prepared fried rice ingredients, and found prominent hydrocolloid accumulation locations as well as typical T_2_ fitting curves [135]. Prepared fried rice, as a heterogeneous type of food, is prone to quality deterioration due to uneven moisture after reheating and irregularities in the oil used by producers. The quick, non-intrusive, and affordable LF-NMR approach offers a potent tool for the real-time quality monitoring of various foods, including prepared fried rice.

## 4. Conclusions and Future Trends

Prepared fried rice is produced through a central kitchen, appropriately packaged to maintain a sterile environment and stored and transported at −18 °C using rapid freezing technologies. However, the use of metamorphic ingredients, low standard production processes, high oil and salt content, and unsafe packaging materials lead to poor quality of prepared fried rice. Compared with traditional processing technologies, irradiation technology and high-voltage electric field technology have the potential to control the microorganisms in prepared fried rice. Microwave, radio frequency and ohmic heating have the potential to sterilize and reheat prepared fried rice. These processing technologies consume less energy and are friendly to the environment. However, these technologies still have limitations. Irradiation and high-voltage electric field sterilization technology require high-capital investment. Microwave, RF, and ohmic heating normally need to be combined with other techniques to improve the uniformity of heating. The processing technologies will develop towards a greener, low-carbon, and environment-friendly direction.

Raman spectral imaging technology, NIR, and LF-NMR, as the new analytical procedures for monitoring, have some advantages. They can non-destructively and efficiently monitor the moisture, protein, and lipid content in food products; and detect quality changes such as color and pH value. These technologies also improve the detection efficiency of enterprises and law-enforcement officers. However, they also have some limitations. The sensitivity of portable Raman detectors needs to be improved. The accuracy of NIR is readily affected by environmental factors such as temperature, which requires correction or compensation to ensure the accuracy and reliability of the data. LF-NMR requires a high-liquid fraction in the food sample to be able to generate a detection signal. Factors such as the viscosity and temperature of the sample may also affect the accuracy. For thicker or multilayer samples, these detection techniques may need to be combined with other techniques to obtain accurate results. With the development of machine learning and artificial intelligence, data processing and model building for Raman spectral imaging technology, NIR, and LF-NMR will be available in the industry. This will significantly improve the accuracy and efficiency in monitoring the quality of prepared fried rice and ensure the safety and consistency of ingredients. The emergence of portable equipment will make it easier to perform rapid testing and analysis of prepared fried rice in the field or at sales terminals. In the future, the development of detecting technology will require the improvement of appropriate software. This allows food researchers to operate equipment and obtain experimental data more easily. Depending on the characteristics of the detection technology, a comprehensive database for a wide range of food products will also need to be established.

In addition, with the rapid development of prepared foods in the market, the relevant laws and regulations should be in place. The raw material standards, processing flows, and product standards (physical and chemical indicators, sensory quality, microbiological indicators, etc.) should be established. Although there are few examples of direct application of processing and testing techniques to prepared fried rice, there have been many studies on the application to individual components of fried rice. In summary, the market of prepared fried rice is expanding, and emerging processing and detection technologies have great potential to be applied to prepared fried rice to ensure a high standard of food quality and safety.

## Figures and Tables

**Figure 1 foods-12-03052-f001:**
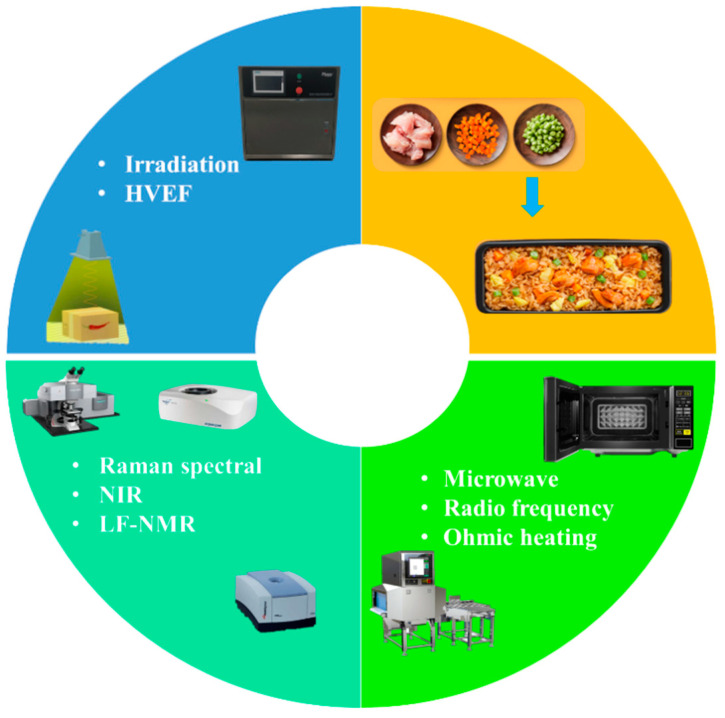
Schematic diagram of processing and quality testing technologies for prepared fried rice.

**Figure 2 foods-12-03052-f002:**
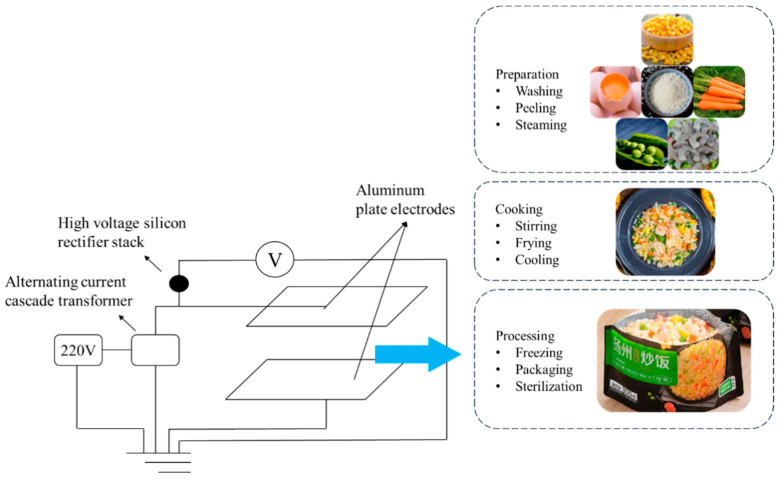
A processing diagram of prepared fried rice under high-voltage electrostatic field, adapted from [36], with permission from ELSEVIER, 2023.

**Figure 3 foods-12-03052-f003:**
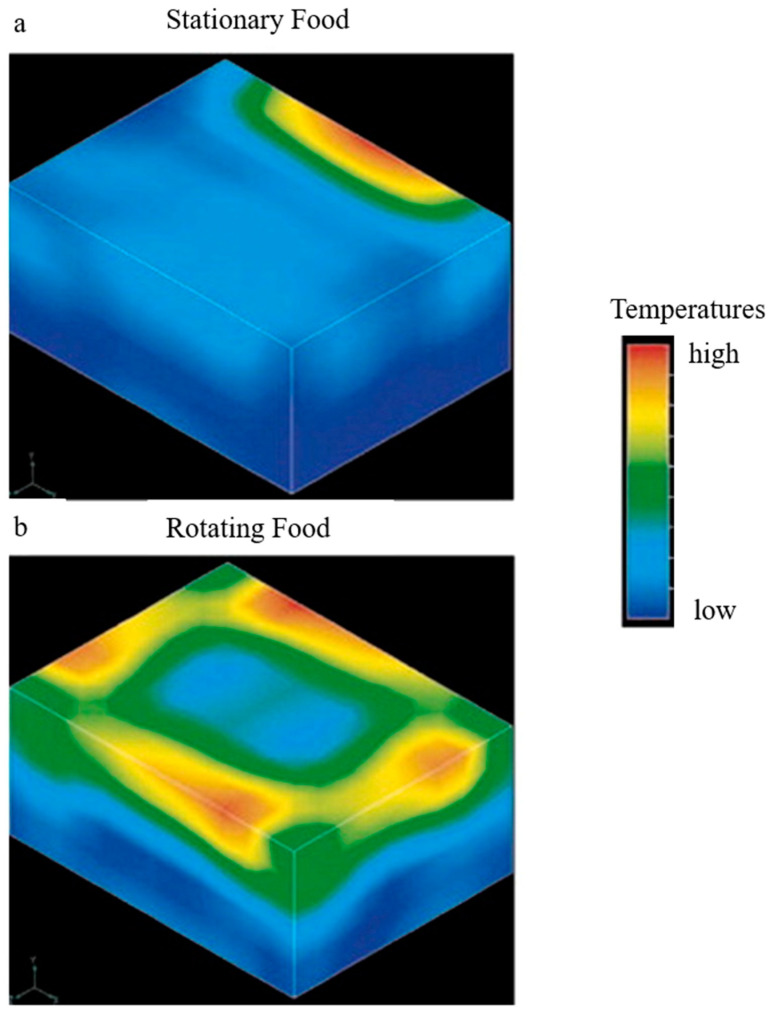
Microwave heating temperature distribution in a non-rotation (**a**) and rotation mode (**b**), adapted from [48], with permission from ELSEVIER, 2023.

**Figure 4 foods-12-03052-f004:**
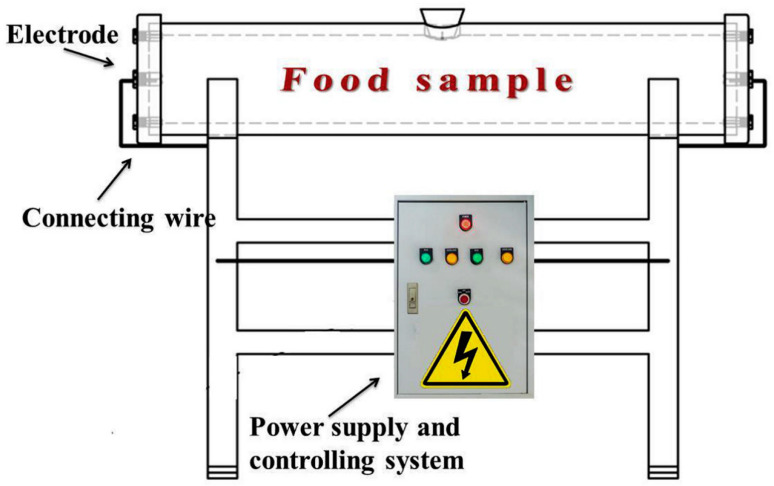
An ohmic food processing system, adapted from [65], with permission from ELSEVIER, 2023 (Gavahian et al., 2019).

**Figure 5 foods-12-03052-f005:**
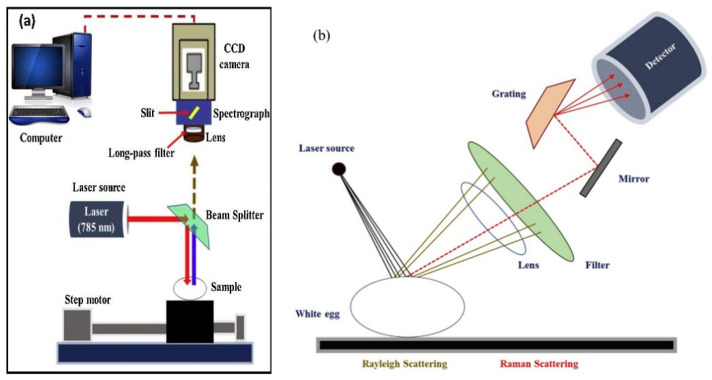
Schematic diagram (**a**) and details (**b**) of Raman hyperspectral imaging system used for measuring real and fake egg samples, adapted from [100], with permission from ELSEVIER, 2023 (Joshi et al., 2020).

**Figure 6 foods-12-03052-f006:**
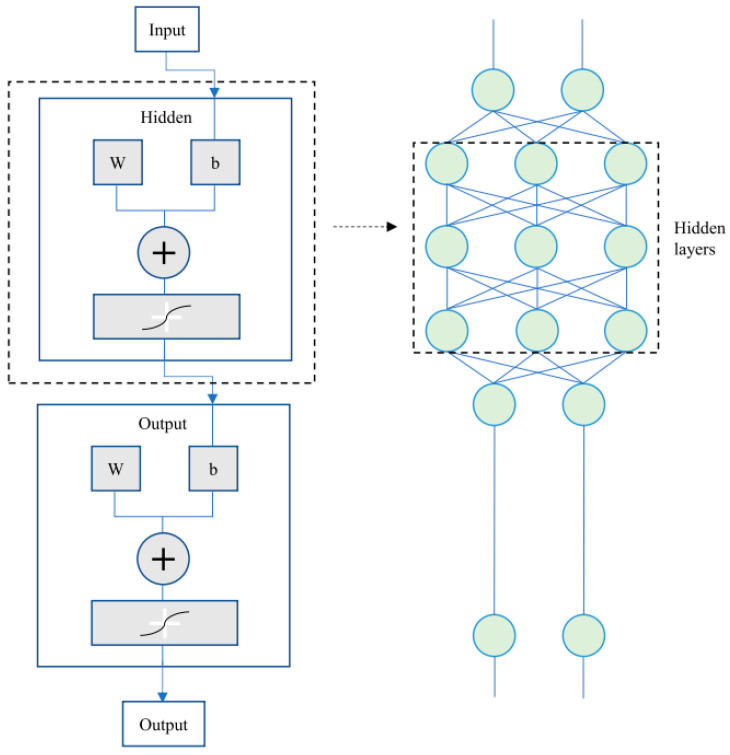
Schematic diagrams of a back-propagation artificial neural network (BP-ANN), adapted from [118], with permission from ELSEVIER, 2023 (Jiang et al., 2023).

**Table 1 foods-12-03052-t001:** Selective prepared fried rice brands and products on the Chinese market.

Brands	Introduction	Representative Products	Features	Main Ingredients	Official Website	Other Information (Reprieved from https://nlc.chinanutri.cn/fq/, accessed on: 8 July 2023)
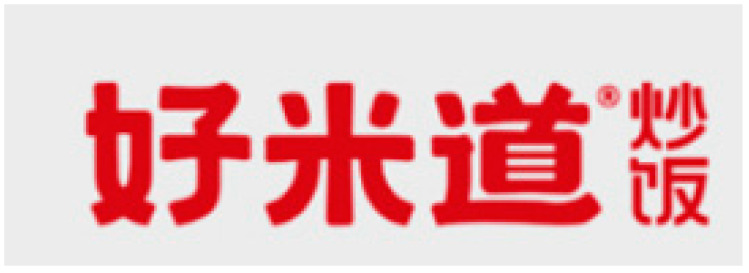	Haitong Foods Founded in 1985, Haitong Foods Group Co., Ltd., Cixi, China. Green food research and development Food production freezing and refrigeration of fruits and vegetables and aquatic products	Mixed fried rice	Precise temperature control Keep fresh at −35 °C	Preserved pork Ox tripe Ham Bacon	https://www.haitonggroup.com/, accessed on: 8 July 2023	Storage conditions and shelf life: −18 °C or 4 °C, 3–12 months Additives: modified soy phospholipids, condiment, pectin, acacia senegal, sodium bicarbonate Capacity: 500–1000 kJ/100 g Protein: 3–5 g/100 g Mainly from rice, eggs, meat, and beans Fat content: 5–15 g/100 g Carbohydrate content: 20–40 g/100 g, mainly from rice and vegetables Sodium content: 400–500 mg/100 g Taste: similar to fresh fried rice; salty, spicy, or sweet and spicy Flavor: emits a strong aroma after heating
		Korean kimchi grilled sausage fried rice		Sausage Pickle		
		Japanese eel fried rice		Eel		
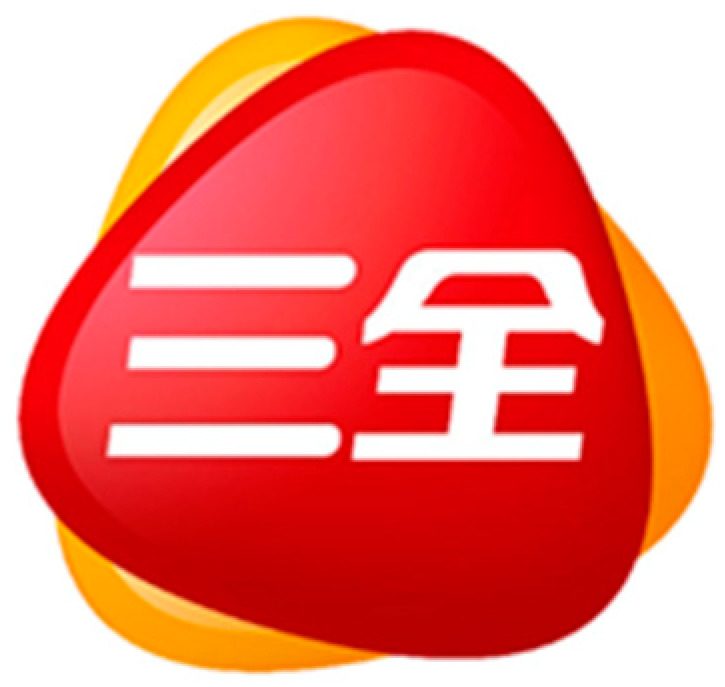	Sanquan Foods One of the companies producing frozen foods in China, which was founded in 1992. Prepared fried rice is a new type of prepared food produced by Sanquan.	Yangzhou fried rice	Don’t need to thaw Don’t need to open the bag Microwave for 3 min in bag	Egg Green bean Corn Carrot Shrimp meat	http://www.sanquan.com/sanquan, accessed on: 8 July 2023	
		Cantonese fried rice		Sausage Green vegetable		
		Thai fried rice		Seafood Pineapple		
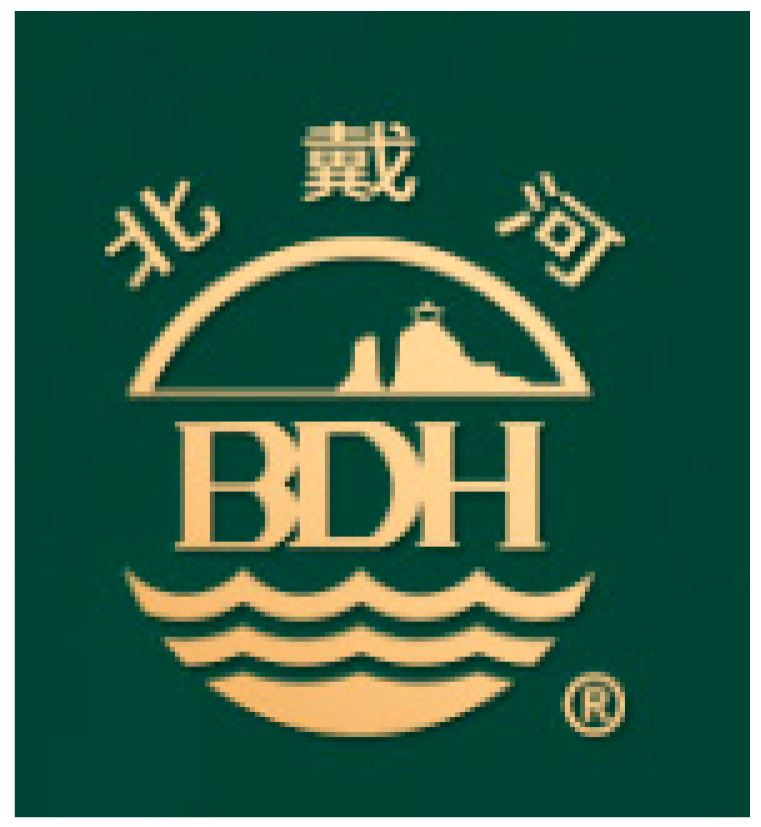	HUANGDAO OCEAN FOOD Founded in 1960, and is a comprehensive food processing enterprise integrating R&D, production, and sales. Canned food Compressed dry food Self-heating food Ready-to-eat food Soup products	Mixed fried rice	Unplugged No fire Heated for 15 min to eat Easy to carry Heated with water Convenient and fast Separated for easy storage	Pork Green bean Carrot Mushroom	http://www.hysp1960.com/, accessed on: 8 July 2023	
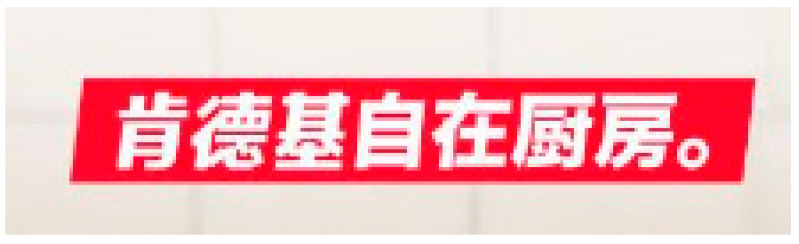	Kentucky Fried Chicken Produced prepared foods since 2020 Products stored in refrigerated cabinets Steak Fried rice/noodles Coffee Chicken soup Chicken breast	New Orleans-style chicken fried rice	Continuous stir-frying above 200 °C Quick-freezing process No firing No washing dishes Microwave heating	Chicken Carrot Green bean	http://www.yumchina.com/Brand, accessed on: 8 July 2023	
		Sichuan tender beef Japanese fried rice		Beef Green bean Carrot		
		Laver salmon fried rice		Salmon Nori Garlic sprouts		
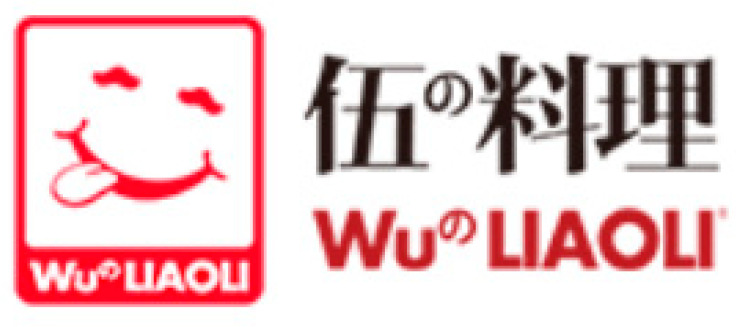	Wu Cuisine is a brand of Wu Group in Quanzhou, China. Wu Group is a catering enterprise integrating the development and production of frozen seasoning kits, brand chain restaurants, and meal hosting with a “central kitchen factory”. Chinese and western frozen seasoning packages Quick frozen food	European-style country beef fried rice	Dishes go from 98 °C to −35 °C in 25 min	Beef Green bean	http://www.wusjt.com, accessed on: 8 July 2023	
		Bacon fried rice		Corn Green bean Bacon		
		Tuna fried rice		Tuna Corn		

**Table 2 foods-12-03052-t002:** Application of Raman spectral imaging technology to the detection of harmful substances in food, adapted from [101], with permission from ELSEVIER, 2023 (Sun et al., 2022).

Analytes		Excitation Laser and Spectral Range	Results	References
Microorganisms	Viruses: influenza AH1N1 virus and human adenovirus	785 nm 1100–1600 cm^−1^	LOD_H1N1_ = 50 pfu/mL LOD_HAdV_ = 50 pfu/mL	[102]
	Bacteria: *Escherichia coli* (*E. coli* ATCC 25922 and K-12 strains) on culture plate	633 nm 600–1800 cm^−1^	LOD = 6 × 10^4^ CFU/mL	[103]
	Fungi: *Aspergillus flavus* (AF36, AF13) infect corn kernels	785 nm 103–2831 cm^−1^	All prediction accuracy ≥ 75.55%	[103]
Mycotoxins	AFB1 in peanut extracts	785 nm 400–1800 cm^−1^	LOD = 0.5μg/L	[103]
Biotoxins	Saxitoxin (STX)	785 nm 400–2000 cm^−1^	LOD = 1 × 10^−7^ M	[104]
Pesticides	Carbamate: thiram in apple juice	785 nm 500–1600 cm^−1^	LOD = 86.1μg/L	[105]
Food authenticity	Authenticating Australian grain-fed and grass-fed beef products	785 nm 600–1900 cm^−1^	All discrimination accuracy ≥ 83%	[105]
Metal ions	Hg^2+^ in natural ground and lake water	785 nm 200–1800 cm^−1^	LOD = 0.1 nm	[106]
Drug residues	Tetracycline in water	532 nm 200–1800 cm^−1^	LOD = 10^−9^ m	[106]
	Malachite green in grass carp, bream fish, and crucian fillets	633 nm 400–1800 cm^−1^	LOD = 0.5μg/L	[107]
	Clenbuterol in pork, chicken, and sausage	785 nm 400–1600 cm^−1^	LOD = 0.05 ng/mL	[108]

## Data Availability

Not applicable.

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
