# Peer review of "Perspectives on Novel Technologies of Processing and Monitoring the Safety and Quality of Prepared Food Products"

_foods, 2023, doi:10.3390/foods12163052_

Round 1

Reviewer 1 Report (Previous Reviewer 1)

Dear authors,

the manuscript has been revised and improved. No more suggestions from my side, the manuscript can be accepted.

Minor spelling.

Author Response

Thank you very much. We have improved some spelling accordingly.

Reviewer 2 Report (Previous Reviewer 4)

The topic of the manuscript is very interesting; however, it shows several drawbacks which must be attended to improve the overall quality of the manuscript.

The title must be modified to “Perspectives on novel technologies of processing and monitoring the safety and quality of ready-to-eat food products” since most of the manuscript focused in different types of ready-to-eat products, including the fried rice.

The sections of the Packaging and Starch modifiers must be removed.

In the Figure 3, it would be desirable to indicate the type of product used for the simulation or analysis, as well as the conditions used.

Regarding to the new analytical procedures for monitoring, it is necessary to include how these techniques have been used to evaluate the quality of different ready-to-eat products as well as the advantages and some limitations. 

Conclusions must be rewritten considering the all the modifications 

It is OK

Author Response

  1. The title must be modified to “Perspectives on novel technologies of processing and monitoring the safety and quality of ready-to-eat food products” since most of the manuscript focused in different types of ready-to-eat products, including the fried rice.

Response: Thank you for your suggestion. We have revised the title to "Perspectives on novel technologies of processing and monitoring the safety and quality of prepared food products". Considering the nature of prepared fried rice that needs to be reheated after the consumer has purchased it, the "prepared food products" may be more accurate than "ready-to-eat food products".

  1. The sections of the Packaging and Starch modifiers must be removed.

Response: Thank you for your suggestion. We have removed the sections of the Packaging and Starch modifiers.

  1. In the Figure 3, it would be desirable to indicate the type of product used for the simulation or analysis, as well as the conditions used.

Response: Thank you for your suggestion. We have revised it.

Lines 284-290: Geedipalli, Rakesh, and Datta (2007) placed 3.6 × 4.7 × 2.1 cm potato cubes in the center of a turntable inside a 2459 Hz microwave oven and heated for 35 s by completing a rotational cycle every 10 s. Figure 3 compared the effect of the same oven with and without the turntable on the uniformity of potato temperature distribution. The static-heated potatoes had concentrated high-temperature areas, while the potatoes heated by the combined microwave rotary heating had more uniform temperature contours on their surfaces and were heated more efficiently with low quality loss.

  1. Regarding to the new analytical procedures for monitoring, it is necessary to include how these techniques have been used to evaluate the quality of different ready-to-eat products as well as the advantages and some limitations.  

Response: Thank you for your suggestion. We have revised it.

Lines 490-501: Insertion of the aggregating agent into the metal nanoparticle aggregates to generate more hotspots, which theoretically produces high SERS surface resonance to further enhance the Raman signal (Ye et al., 2023). Cheng et al. (2019) combining QuECHERS extraction and SERS and Ye et al. (2023) using AgNP substrate-assisted SERS technique with 0.5 M NaCl as the agglomerating agent, both demonstrated that Raman spectroscopy can rapidly detect acrylamide in fried foods and the results are consistent with LC-MS/MS, which can be used as a technical tool for on-site screening. It is worth noting that the SERS technique has limitations. The experimental conditions and control parameters need to be optimized because of various samples with different surface structure and preparation. It currently only used in typical fried foods such as potato chips, needing to further expand the scope of application.

Lines 507-516: This technique does not require complex pretreatment of prepared fried rice, is simple to operate, and combines with fingerprinting to identify multiple molecules simultaneously for effectively regulate the safety of fried rice ingredients (Zhang et al., 2020), such as rapid online monitoring or on-site detection of pork ractopamine and clenbuterol hydrochloride residues using SERS with chemometric methods (Zhao et al., 2017), non-destructive detection of fungal spore counts and textural features of corn (Long et al., 2022), distinguishing of aflatoxin contaminated corn kernels (Tao et al., 2021), and classification of the fake and real eggs (Figure 6) (Joshi et al., 2020).

Lines 582-586: Peyvasteh et al. (2020) used NIR coupled with PCA to determine the relative absorbance changes of oxidized myosin and myoglobin, fat, water, and collagen in the visible spectrum of pork for rapid evaluation of pork freshness. These ingredients are common sources of protein in prepared fried rice and directly affect its quality.

Lines 621-628: Zhang, Zhang, and Mujumdar (2021) concluded that ice crystals formed after freezing cause damage to the structure of muscle cells, leading to migration of water molecules. Water loss in chicken breast, beef and seafood, lipid types on fermented sausage and myofibrillar protein gels can be sensitively detected by LF-NMR technique. Moreover, the integrity of proteins is revealed by the decrease in the lateral relaxation time of their protons due to protein aggregation, which further leads to the decrease in the lateral relaxation time of water protons.

Lines 669-680: Raman spectral imaging technology, NIR and LF-NMR, as the new analytical procedures for monitoring, have some advantages. They can non-destructively and efficiently monitor the moisture, protein and lipid content in food products, and detect quality changes such as color and pH value. These technologies also improve the detection efficiency of enterprises and law enforcement officers. However, they also have some limitations. The sensitivity of portable Raman detectors needs to be improved. The accuracy of NIR is readily affected by environmental factors such as temperature, which requires correction or compensation to ensure the accuracy and reliability of the data. LF-NMR requires a high liquid fraction in the food sample to be able to generate a detection signal. Factors such as the viscosity and temperature of the sample may also affect the accuracy. For thicker or multilayer samples, these detection techniques may need to be combined with other techniques to obtain accurate results.

5.Conclusions must be rewritten considering the all the modifications.

Response: Thank you for your suggestion. We have revised it.

Lines 669-690: Raman spectral imaging technology, NIR and LF-NMR, as the new analytical procedures for monitoring, have some advantages. They can non-destructively and efficiently monitor the moisture, protein and lipid content in food products, and detect quality changes such as color and pH value. These technologies also improve the detection efficiency of enterprises and law enforcement officers. However, they also have some limitations. The sensitivity of portable Raman detectors needs to be improved. The accuracy of NIR is readily affected by environmental factors such as temperature, which requires correction or compensation to ensure the accuracy and reliability of the data. LF-NMR requires a high liquid fraction in the food sample to be able to generate a detection signal. Factors such as the viscosity and temperature of the sample may also affect the accuracy. For thicker or multilayer samples, these detection techniques may need to be combined with other techniques to obtain accurate results. With the development of machine learning and artificial intelligence, data processing and model building for Raman spectral imaging technology, NIR and LF-NMR will be available in the industry. This will significantly improve the accuracy and efficiency in monitoring the quality of prepared fried rice and ensure the safety and consistency of ingredients. The emergence of portable equipment will make it easier to perform rapid testing and analysis of prepared fried rice in the field or at sales terminals. In the future, the development of detecting technology will require the improvement of appropriate software. This allows food researchers to operate equipment and obtain experimental data more easily. Depending on the characteristics of the detection technology, a comprehensive database for a wide range of food products will also need to be established.

Reviewer 3 Report (New Reviewer)

To me, the article can be acceptable.  

Minor editing of English language required.

Author Response

Thank you very much. We have checked the language carefully.

Round 2

Reviewer 2 Report (Previous Reviewer 4)

Authors have attended all suggestion in a good way, improving the overall quality of the manuscript.

It is ok

This manuscript is a resubmission of an earlier submission. The following is a list of the peer review reports and author responses from that submission.

Round 1

Reviewer 1 Report

Dear Authors,

the present manuscript describes an interesting state-of-the art related to technologies of processing and quality detection for prepared fried rice. Where is the novelty and why now?

Lines 35-38: please refer to China market, otherwise give some worldwide examples

Lines 47-54: references are missing to support the statements, please add relevant references

Table 1 references missing, please cite for each value a related reference

Figure 4 is related to lasagna. Lines 462-463: "Prepared fried rice and lasagna have similar compositions and both require cold chain circulation and quality maintenance." - still do not know if proper to show the mechanism for lasagna and to assume looks the same for fried rice. Please reformulate. Reference to support these statements.

Lines 472-479: please add references

Figure 5 - does not have any relevance to fried rice, and the whole section 3.1. Raman spectroscopy imaging technology -does not show any information related for fried rice

In general, 3. New testing technologies in prepared fried rice quality and safety seems to be a general section describing the techniques but not so much woth evifdance and examples related to fried rice. Please add some relevant information.

The quality of English is fine.

Reviewer 2 Report

The paper proposes a literature review and update on fried rice. The paper deals with general aspects of the preparation and preservation of fried rice. It then presents the effects of the use of some emerging technologies to improve its processes and finally addresses aspects on modern and non-destructive analysis to evaluate the quality and composition of fried rice dishes.

The article is interesting and well presented, however there are opportunities for improvement, which are mentioned below:

- The relevance of table 1 needs to be reviewed, as it has only one row.

- Table 2 could be improved: the row boundaries are not clearly visible, and the product brands are all in Chinese so that the reader outside China will not understand them, the English translation could be placed below the respective brand logos.

- Revise the wording of l59 where hazardous chemicals produced by microbial metabolism are mentioned.

- In l92 it talks about the taste of fried chicken, this part is not well understood.

Review what is the contribution of figure 2.

In l143 aflatoxins are mentioned as microorganisms, review and correct.

In l171 it is missing what type of radiation it refers to (gamma radiation, x?).

In l188 it is mentioned that high voltage electric fiel applies to fried foods, revise the assertion because it does not necessarily apply to fluid foods.

In l206 it talks about freeze concentration, does it refer to cryoconcentration?

The diagram in figure 2 can be improved.

It is striking that there is no mention of ohmic heating as an emerging technology available for heating food.

Figure 4 is too small, consider modifying it to improve the understanding of what you want to show.

In l 327 what is meant by digestive properties, improve the wording.

l332 prefer the term retorting to autoclaving.

in L348 delete the sentence about what is in the prepared rice.

l442 what is meant by changing an electron, it is not clear.

in l467 it is not clear what is meant by t22 peak time.

l473 it is not clear to this reviewer how microorganisms produce acrylamides.

Check the contribution of figure 5 and if 5b is really a photograph?

Reviewer 3 Report

I am sorry to say that the manuscript lacks cohesion, the title talks about fried rice, but very different foods are included. It seems a succession of references without a specific purpose.

The figures bear little relation to the text (eg figure 3).

Where does the data in Table 1 come from?

The references are a chaos, their numbering or the authors do not follow any order, there are repeated ones, it seems that there is a significant number of self-citations,...

Reviewer 4 Report

The research work about the application of novel technologies for food processing and for quality assurance is very interesting. However, the current manuscript has several drawbacks which must be attended. Firstly, it is necessary to add a section describing the processing of fried rice, including the critical points of processing. Likewise, it is important to indicate the processing steps where is included the new technology as well as the possible advantages. Then, although, most of the novel food processing technologies are very well described, even some examples about the effect of these technologies in different products, there is not enough information about the application and/or impact in the fried rice processing. In fact, most of the application of the novel technologies focused in the inactivation of pathogens but it is unclear its relationship with the fried rice processing. Therefore, all subsections, either for processing or for quality test, must be include the more studies performed on the processing of fried rice and the advantages obtained. Finally, the conclusions must be rewrite carefully based on results of the use or application of these technologies in the processing of fried rice. The actual conclusion section is a brief summary of the possible advantages if novel technologies will be applied for prepared fried rice but these are not based on experimental results.

It's Ok